# Pigmented Purpuric Dermatoses: A Complete Narrative Review

**DOI:** 10.3390/jcm10112283

**Published:** 2021-05-25

**Authors:** Cristina B. Spigariolo, Serena Giacalone, Gianluca Nazzaro

**Affiliations:** 1Dermatology Unit, Fondazione IRCCS Ca’ Granda Ospedale Maggiore Policlinico, 20122 Milan, Italy; cristina.spigariolo@gmail.com (C.B.S.); serenagiacalone92@gmail.com (S.G.); 2Department of Pathophysiology and Transplantation, Università degli Studi di Milano, 20122 Milan, Italy

**Keywords:** pigmented purpuric dermatosis, capillaritis, Schamberg disease, lichen aureus, purpura of Majocchi, purpura of Doucas and Kapetanakis, dermatitis of Gougerot and Blum

## Abstract

Pigmented purpuric dermatoses (PPD) include several skin diseases characterized by multiple petechial hemorrhage as consequence of capillaritis. PPD generally present with red to purple macules that progressively evolve to golden-brown color as the hemosiderin is reabsorbed. These lesions, often asymptomatic or associated with mild pruritus, usually occur on the lower extremities and may be a diagnostic and therapeutic challenge both for general practitioners and specialists in internal medicine or flebology. Clinical presentations include many subtypes that have been described over the years, although histology is usually superimposable. Prompt recognition and patient reassurance on the benign nature of these diseases is crucial. In this comprehensive review, we focused on pathogenesis and clinical pictures.

## 1. Introduction

Pigmented purpuric dermatoses (PPD) represent a group of cutaneous diseases characterized by petechial hemorrhage as a consequence of capillaritis [1]. Extravasated erythrocytes result in purpura, and hemosiderin-laden macrophages give a red–brown appearance to older lesions. PPD usually present as remitting-relapsing non-palpable flat purpura with bilateral distribution on the legs of elderly. Although adults are more frequently affected, there are numerous reports of childhood PPD in literature [2]. These conditions have no systemic findings, but occasionally lead to a patient evaluation to exclude thrombocytopenia or vasculitis because of the purpuric (usually petechial) nature of the lesions and clinical misdiagnosis. Based on the clinical appearance, PPD are classified as five major entities: progressive pigmentary dermatosis or Schamberg disease (the commonest), pigmented purpuric lichenoid dermatosis of Gougerot and Blum, purpura annularis telangiectodes of Majocchi, eczematid-like purpura of Doucas and Kapetanakis, and lichen aureus [3]. Other forms are rare and with an unusual presentation [1].

## 2. Aetiology and Pathogenesis

The aetiology is unknown but physical activity; venous hypertension; capillary fragility, especially in lower limbs; and local infections seem to play a role in the pathogenesis [1]. Some drugs, such as acetaminophen, non-steroidal anti-inflammatory drugs, antibiotic, and oral antidiabetic (Table 1), may appear as a trigger, in particular in Schamberg disease [4,5,6]. PPD can be associated with systemic diseases [1], such as diabetes mellitus, rheumatoid arthritis, systemic lupus erythematosus, thyroid dysfunction, mycosis fungoides, hematological and solid neoplasms, liver disease, and hyperlipidemia; however, most of the PPD remain idiopathic. All PPD result from minimal inflammation and hemorrhage of superficial papillary dermal vessels, usually capillaries. The reason for inflammation is unknown, and there is no association with any abnormality of coagulation. Immunopathologic studies suggest that the vascular damage and erythrocyte extravasation are secondary to a localized cell-mediated immunologic event [7]. Some studies suggest an upregulation of cell adhesion molecules, such as LFA-1 (lymphocyte function antigen-1) and ICAM-1 (intercellular adhesion molecule-1) in inflammatory cells and ICAM-1 and ELAM-1 (endothelial leukocyte adhesion molecule-1) in endothelial cells [8].

## 3. Pathology

Despite varying clinical features, all PPD share similar histopathology. The four characterizing elements are dilated blood vessels, with endothelial cell swelling, deposit of hemosiderin, red cell extravasation, and perivascular lymphocytic-macrophage infiltration. It is also possible to define the pathological patterns based on the predominant inflammatory presentation: spongiotic, interface/lichenoid, perivascular, or granulomatous pattern (Figure 1a) [9]. Epidermal spongiosis and lymphocytic exocytosis are common in all PPD but are more pronounced in eczematid-like purpura of Doucas and Kapetanakis. Lichen aureus (Figure 1b) is characterized by a band-like dermal infiltrate separate from normal epidermis by grenz zone. Lichenoid infiltrate and spongiosis with patchy parakeratosis are the distinguishing features of dermatosis of Gougerot and Blum. A rare granulomatous form, variant of a chronic pigmented purpura, has also been described [10].

## 4. Clinical Variants

### 4.1. Schamberg Disease or Progressive Purpuric Pigmented Dermatosis

Schamberg disease, first described in 1901 [11], is the most common form of PPD [1]. It may affect all ages but commonly occurs in middle-aged to older men and less frequently children [2,12]. Typical localizations are lower limbs but can also involve thighs, buttocks, trunk, and arms. Clinical examination usually shows pinpoint petechiae, likened to “cayenne pepper”, that flows in yellow-orange-brown patches with an oval to irregular outline (Figure 2). It is a chronic-persistent dermatosis that, in some cases, can extend to the proximal areas. Most patients are asymptomatic, whereas some complain of mild itching. Venous insufficiency might play a role in their localization. Moreover, chronic alcohol uptake seems to be related to the development of purpura and Schamberg disease, especially when liver damage is associated [13].

### 4.2. Purpura Annularis Telangiectodes of Majocchi 

Purpura annularis telangiectodes of Majocchi was first described in 1896 [11]. It is an uncommon PPD that mainly affects adolescents and young adults, especially women [14]. Early lesions are bluish to red annular macules in which dark-red telangiectatic puncta appear; subsequently, they have a centrifugal extension with central progressive resolution, and slight atrophy, giving them an annular configuration (Figure 3). The eruption begins bilaterally on the lower limbs and then extends to the upper extremities, but it is also described as a rare unilateral form [15]. Lesions may vary from few in number to innumerable; patients can be asymptomatic or complain of mild itch or burning. As far as the trigger factors, in addition to those already mentioned above, there is sclerotherapy due to direct endothelial damage with resultant red blood cell extravasation and hemosiderin deposition [16]. Touraine [17] described the “arciform telangiectatic variant” characterized by a less number of lesions with lower size and arciform and irregular edges.

### 4.3. Pigmented Purpuric Lichenoid Dermatitis of Gougerot and Blum 

Gougerot and Blum defined this entity in 1925 [11]. It is a rare form characterized by two type of lesions [18]: purpuric red–brown lichenoid papules and Schamberg-like purpura (Figure 4). Lesions are usually located on limbs and not uncommonly involved oral mucosa [19]. This disease usually affects middle-aged to older men, has a chronic course, and occasionally presents with pruritus. However, PPD of Gougerot-Blum might clinically resemble Kaposi’s sarcoma, mycosis fungoides, cutaneous vasculitis, and traumatic purpura.

### 4.4. Lichen Aureus

First described as lichen purpuricus in 1958, it was successively named lichen aureus by Calnan in 1960 [11]. It is a more localized, persistent, intensely purpuric eruption, rarely painful [20]. It appears as a purpuric lesion associated with lichenoid papule, usually located on the lower extremities, often over a perforator vein, occasionally on the trunk, upper extremities, buttocks, or face (Figure 5). The color varies from golden to rust to purple–brown and justifies the term “lichen aureus” as well as the pathologic pattern of a lichenoid dermatitis. Unlike with other forms, young men are the most affected.

### 4.5. Eczematid-Like Purpura of Doucas and Kapetanakis 

This form is considered an inflammatory and more extensive variety of Schamberg disease, presented for the first time in 1953 [11]. In fact, it tends to spread cranially from lower limb up to the abdomen, and it is characterized by intense itching and scaly petechial or purpuric macules, papules, and patches [21] (Figure 6). In confirmation of this, pathology shows spongiosis and intense lymphocytic perivascular infiltration. Resolution is always spontaneous but relapses are frequent. Association with allergic contact dermatitis to rubber and clothing has been described.

### 4.6. Other Forms

Granulomatous pigmented purpura is a rare variety of PPD that seems to affect Asian people more frequently than Caucasians. It was first described in 1997 by Saito et al. [22]. It generally occurs in middle-aged to elderly patients and commonly follows a chronic waxing and waning course. A correlation with dyslipidemia was reported in literature [23]. Clinical examination often shows brown patches with superimposed hemorrhagic papules on the lower legs. Histologic specimen is characterized by hemorrhagic granulomatous infiltrate, including histiocytes, located in the upper dermis.

Linear pigmented purpura is an uncommon form that can occur in children and adolescence. The terms quadrantic capillaropathy and unilateral linear capillaritis have been used to describe this entity in the past [24]. It is characterized by lesions that are similar in appearance to lichen aureus or Schamberg disease but are unilateral and linear [11,15]. It must be differentiated from other dermatoses such as lichen aureus (segmental variant), unilateral nevoid telangiectasia, and angioma serpiginosum, which have linear aspects such as linear purpura.

Itching purpura of Loewenthal should be considered as a symptomatic variant of Schamberg disease that affects almost exclusively adults [25]. It is characterized by abrupt onset and severe itching on lower extremities.

## 5. Differential Diagnosis

Clinical presentation is always sufficient for diagnosis (Table 2). PPD must be differentiated from purpura caused by platelet alteration and non-thrombocytopenic purpura. 

Platelet alteration could result from thrombocytopenic, thrombocytopatic, or thrombocytosis syndromes. These forms are usually associated with bleeding in other locations, or in case of thrombocytosis with a prothrombotic state. Thus blood count test and peripheral blood smear could help in doubtful cases. On the contrary, there is no platelet alteration in non-thrombocytopenic purpura. Some examples are “exercise-induced purpura”, which occurs on lower legs after unusual or major muscular activity [26], or “solar purpura”, which appears acutely after sun exposure [27]. Pigmented purpuric eruptions on the legs must be also differentiated from dermal hemorrhage secondary to venous hypertension, which presents with petechiae superimposed on diffuse hemosiderosis, especially in elderly people with stasis dermatitis. 

Hypergammaglobulinemic purpura of Waldenström is characterized by hypergammaglobulinemia; recurring purpura, especially on the legs; elevated erythrocyte sedimentation rate; and the presence of rheumatoid factor indicative of circulating immune complexes. It is often preceded by slight itch, mild tingling, or burning, and it may be aggravated by tight-fitting garments, prolonged standing, and heat [28].

Sometimes biopsy is required to distinguish the lichenoid variant from cutaneous small vessel vasculitis. Leukocytoclastic vasculitis/cutaneous venulitis are characterized by palpable erythematous papules that do not blanch on diascopy.

Finally, purpuric clothing dermatitis, hyperglobulinemic purpura, early mycosis fungoides, stasis pigmentation, scurvy, leukocytoclastic vasculitis, and drug-hypersensitivity reactions can mimic Schamberg disease.

## 6. Diagnosis

Diagnosis is clinical in the majority of cases: orange-red-brown macules with purpuric spots associated with specific features of variants mentioned above are decisive. Doubtful forms benefit from biopsy. All subtypes share dilated blood vessels, endothelial cell swelling, deposit of hemosiderin, red cell extravasation, and perivascular lymphocytic-macrophage infiltration. Moreover, epiluminescence may address the correct diagnosis: coppery-red pigmentation, red dots, and globules, due to the extravasation of red blood cells and to hemosiderin in the histiocytes and dotted vessels, are the main findings. Brown dots and globules result from melanocytes or melanophages at the dermo-epidermal junction [29].

## 7. Investigation

Blood count test and peripheral blood smear are necessary to exclude thrombocytopenia; coagulation screen can help to exclude other possible causes of purpura. Sometimes, tests for rheumatoid factor, anti-nuclear antibodies, anti-Hepatitis B Virus, and anti-Hepatitis C Virus antibodies can be useful to identify purpura associated with autoimmune diseases or secondary to hepatitis.

## 8. Therapy

Standard guidelines of treatment are not available for PPD [3]. They are usually persistent and resistant to therapies. Thus, conservative or non-pharmacologic management may be considered in the case of asymptomatic patients [1]. Therapeutic options include topical and oral drugs, photo-therapy, and laser-therapy (Table 3).

In the pruritic forms or evident erythema, as in purpuric lichenoid dermatitis of Gougerot-Blum, topical corticosteroids, often associated with moisturizing cream, are helpful [30]. Duration of therapy and steroids potency are variable [11,31]. Prolonged steroid use should be discouraged due to the risk of local atrophy. Topical pimecrolimus 0.1% was effectively used in lichen aureus in a child: color improvement was noticed after three months of daily treatment, while a complete resolution after one year [32]. Unlike steroids, calcineurin-inhibitors are not responsible for skin atrophy.

Oral rutoside (50 mg twice a day) and ascorbic acid (500 mg twice a day), through the increase capillary resistance and with antioxidative radical scavenging activities, cleared three patients in a trial within 4 weeks [33,34]. Pentoxifylline at different dosages (300 mg daily [35]; 400 mg bid [36]) has proven to be a satisfactory drug in these conditions. It seems to act at level of T-cell adherence to endothelial cells and keratinocytes. A therapy with daily oral flavonoids (diosmin 450 mg, and hesperidin 50 mg, and euphorbia prostata extract 100 mg) and calcium dobesilate 500 mg successfully treated a case of Schamberg disease in 14 days [37]. Colchicine 0.5 mg twice a day for two months was also adopted successfully by Cavalcante et al. for treating Schamberg disease [38]. Tamaki et al. reported a dramatic improvement within 7–14 days of treatment in five patients receiving 500 to 750 mg griseofulvin daily. Therapeutic response seems to come from local inflammation reduction [39]. The efficacy of oral cyclosporine in chronic PPD is worth mentioning, given that it supports the immune-mediated pathogenesis of some of these forms [40], although, due to the benign nature of PPD and the serious adverse-reaction profile of cyclosporine A, it is not recommended as a first-line option. Other treatments reported in literature included tranexamic acid and heparinoid gel [9]. Associated venous stasis should be treated by compression, and prolonged leg dependency should be avoided [7].

PUVA-treatment has been successfully used in extended forms of Gougerot-Blum dermatitis [41], lichen aureus [42], and Schamberg disease [43]. Other reports suggest the potential role of narrowband UVB [44,45].

Finally, another valid option is laser-therapy. In particular, 595-vascular laser proved to be effective in Schamberg disease [46]. Other reports suggest the positive role of fractional non-ablative 1540 nm erbium with 4 monthly sessions in Schamberg disease [47]. Dye laser was also adopted for treating lichen aureus [48,49].

## 9. Conclusions

PPD include a group of skin disorders characterized by petechial hemorrhage, deposit of hemosiderin, red cell extravasation, and dilated blood vessels. They are classified on the basis of clinical presentation since histopathology is almost superimposable. Sometimes PPD are associated with venous and capillary dysfunction, which explain the frequent involvement of lower legs. Although in the majority of cases they are idiopathic, routine blood tests and coagulation tests are recommended to exclude a possible association with thrombocytopaenia or blood clotting disorder. As far as the treatment, most cases are resistant and usually relapse thus non-pharmacological measures must be always considered. Topic steroids might be helpful for inflammatory and itching variants.

## Figures and Tables

**Figure 1 jcm-10-02283-f001:**
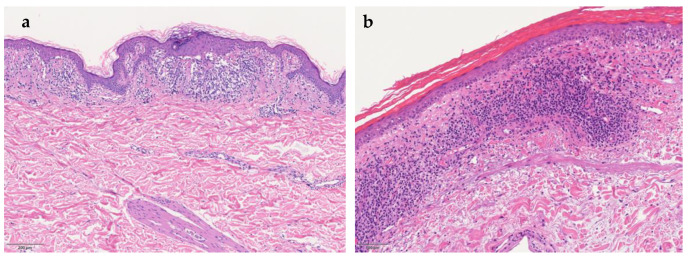
(**a**) Histopathological specimen shows dilated blood vessels, mild perivascular and lichenoid lymphocytic infiltrate, exocytosis of erythrocyte, and focal epidermal spongiosis. (Hematoxylin- eosin stain). (**b**) Lichen aureus: dense lichenoid infiltrate associated with dilatated blood vessels and normal epidermis.

**Figure 2 jcm-10-02283-f002:**
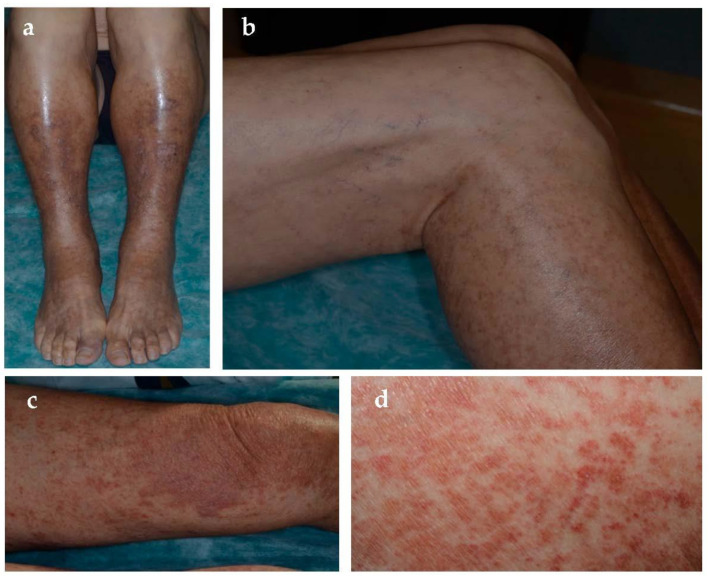
Schamberg disease. (**a**,**b**) Lower leg localization of petechiae superimposed on diffuse hemosiderin deposits in venous hypertension. Proximal extension towards thighs is visible. (**c**,**d**) Discrete yellow-red patches with superimposed petechiae.

**Figure 3 jcm-10-02283-f003:**
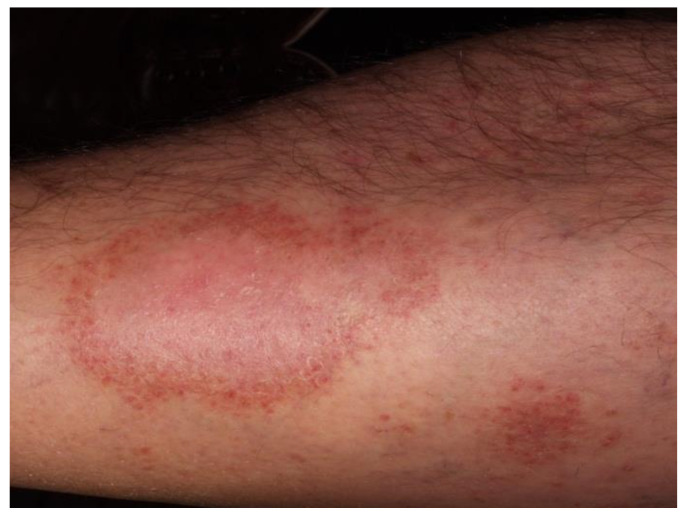
Purpura annularis telangiectodes of Majocchi. Annular plaques with central atrophic area and cayenne pepper petechiae in the border.

**Figure 4 jcm-10-02283-f004:**
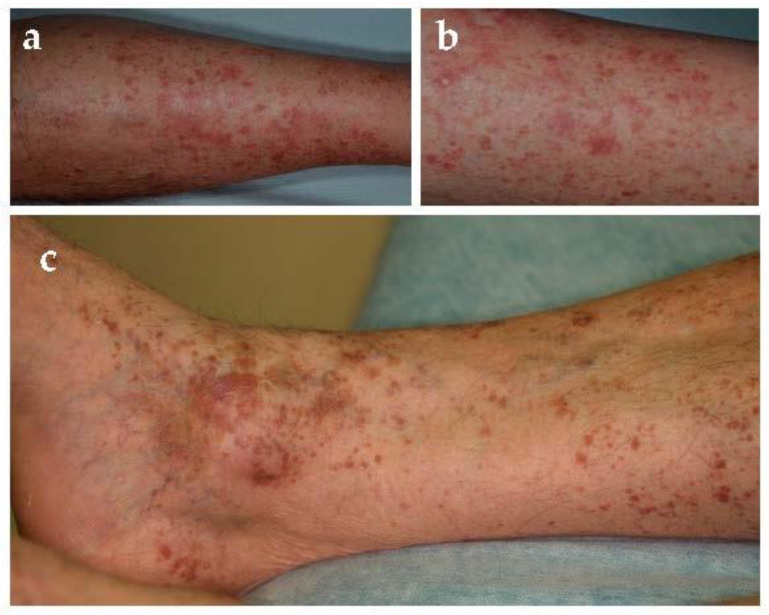
Pigmented purpuric lichenoid dermatitis of Gougerot and Blum. (**a**–**c**). Purpuric red-brown lichenoid papules on lower extremities.

**Figure 5 jcm-10-02283-f005:**
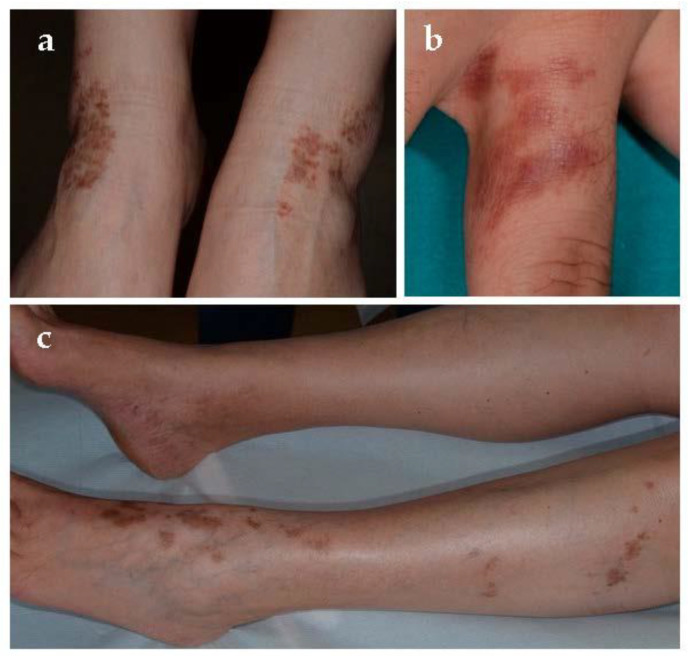
Lichen aureus. (**a**) Symmetrical and bilateral yellow-brown patches involving two malleoli. (**b**) Single localization of a rust-color patch on the first phalanx of the third finger. (**c**) Lichen aureus on the legs.

**Figure 6 jcm-10-02283-f006:**
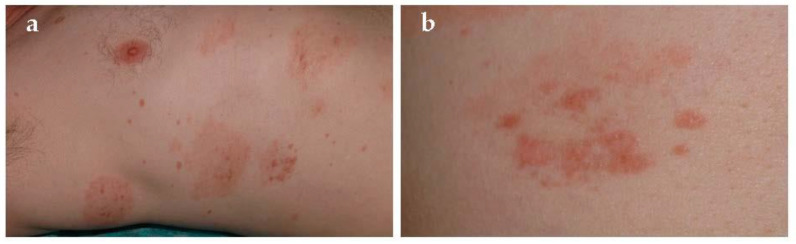
Eczematid-like purpura of Doucas and Kapetanakis. (**a**,**b**) Scaly petechial and purpuric macules and patches involving the abdomen.

**Table 1 jcm-10-02283-t001:** Drugs inducing pigmented purpuric dermatoses.

**Analgesic drugs**
Acetaminophen
Aspirin
**Oral antidiabetic agents**
Glipizide
Glybuzolo
**Antibiotics**
Pefloxacin
**Neurological medications**
Lorazepam
Meprobamate
Carbromal
Chlordiazepoxide
**Cardiovascular medications**
Hydralazine
Dipyridamole
Reserpine
**Others**
Thiamine
Interferon-alfa
Medroxyprogesterone acetate

**Table 2 jcm-10-02283-t002:** Distinctive features of pigmented purpuric dermatosis.

Type	Localization	Clinical Features	Histopathology	Local Therapy	Associations
Schamberg disease	Lower limbs Less frequent: trunk, arms, thighs, or buttocks.	Middle-aged to older men Orange- red macules with pinpoint petechiae, likened to “cayenne pepper” Asymptomatic	Dilated blood vessels Endothelial cell swelling, deposit of hemosiderin Red cell extravasation Perivascular lymphocytic-macrophage infiltration	Moisturizing cream Topical steroids Topical calcineurin Phototerapy	Venous insufficiency Chronic alcohol uptake Drugs Sclerotherapy
Purpura annularis telangiectodes of Majocchi	From lower limbs to the upper extremities	Female adolescents and young adults Centrifugal extension of red annular macules with red telangiectatic puncta with central progressive resolution Asymptomatic or mild itch or burning	Identical to Schamberg disease		
Pigmented purpuric lichenoid dermatitis of Gougerot and Blum	Lower limbs Occasional involvement of the trunk and oral mucosa	Middle-aged to older men Purpuric red–brown lichenoid papules Pruritus	Lichenoid infiltration		
Lichen aureus	Lower limbs, often over a perforator vein Occasionally trunk, upper extremities, buttocks, or face	Young men Localized form: solitary or small in number Lichenoid papules with the tendency to coalesce into plaques Gold color Intense itch	Lichenoid infiltration with Grenz zone Unchanged epidermis Dilated blood vessels Perivascular lecucocyte infiltration		
Eczematid-like purpura of Doucas and Kapetanakis	From head to lower limbs	Inflammatory variant of Schamber disease Intense itch Scaly petechial or purpuric macules, papules and patches	Spongiosis Intense lymphocytic perivascular infiltration Neutrophils		

**Table 3 jcm-10-02283-t003:** Oral therapies used for treatment of pigmented purpuric dermatosis.

Oral Medications	Mechanism of Action	Dosage	(Level of Evidence)	Type of PPD
Rutoside [34]	Antioxidative radical scavengers (control SOCS3 gene expression and STAT3 signaling) Reduce capillary permeability and fragility	50 mg bid (mean treatment duration 8.2 months)	Retrospective two centre studies (II)	Schamberg disease
Ascorbic acid [34]	Antioxidative radical scavengers Essential for collagen synthesis	1000 mg day (mean treatment duration 8.2 months)	Retrospective two centre studies (II)	Schamberg disease
Pentoxifylline [9,35,36]	Reduction of T-cell adherence to endothelial cells and keratinocytes (decreased expression of ICAM-1)	300 day for 8 weeks 400 bid from 4 to 8 weeks	Case report (III) Retrospective review (Ia)	Schamberg disease
Mix of venoactive agents [37] with: Oral flavonoids [37]: Diosmin Hesperidin Euphorbia prostata extract	Decrease capillary permeability Inhibition of oxygen free radical production and lipid peroxidation Decrease synthesis of prostaglandins E2 or F2 and thromboxane B2 Decrease of endothelial activation: reduction of ICAM-1 and VCAM-1 expression Decrease of blood viscosity	Diosmin 450 mg day for 2 weeks Hesperidin 50 mg day for 2 weeks Euphorbia prostata extract 100 mg day for 2 weeks	Case report (III)	Schamberg disease
Calcium dobesilate [37]	Inhibition of prostaglandins and thromboxanes synthesis Vessel relaxation due to production of NO Lowering of blood viscosity Downregulation of VEGF expression	500 mg day for 2 weeks	Case report (III)	Schamberg disease
Colchicine [38]	Immunomodulating function due to blockage of T-cell chemiotaxis	0.5 mg bid for 8 weeks	Case report (III)	Schamberg disease
Griseofulvin [39]	Immunomodulating function due to reduced expression of interferon- 7-induced HLA-DR of keratinocytes	500–750 mg day for 1–2 weeks	Clinical trial (III)	Schamberg disease Eczematid like purpura
Cyclosporine [40]	Immunomodulating function due to decreased function of T-cell	2–5 mg/kg day for 6 weeks	Case report (III)	Schamberg disease

Legend SOCS3 (suppressor of cytokine signalling 3), STAT3 (signal transducer and activator of transcription 3), ICAM-1(intercellular adhesion molecule 1), VCAM-1 (vascular cell adhesion molecule 1), NO (nitric oxide), and VEGF (vascular endhotelial growth factor).

## Data Availability

No new data were created or analyzed in this study. Data sharing is not applicable to this article.

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
