# Peer review of "Pigmented Purpuric Dermatoses: A Complete Narrative Review"

_jcm, 2021, doi:10.3390/jcm10112283_

Round 1
Reviewer 1 Report
This is just a narrative review.
In order to add a more practical use of the information I would suggest to add a table with the clinical features and the key differences among the entities, plus proposed lab exams and therapeutic suggestions
Author Response
Dear Reviewer 1,
thank you for your suggestions. A summary table of pigmented purpuric dermatosis has been added. As far as oral therapy, another table has been inserted in the paper, as suggested by Reviewer 2. On the contrary, since there is no specific exam for every single dermatosis we unfortunately can not specify this data. However the most common associations has been inserted.
Reviewer 2 Report
This is a detailed and well-written review. Thank you for including good-quality pictures.
I have a few minor suggestions:
1. Change "Basing on the clinical appearance" to "Based..."
2. Change "classified in five major entities" to "classified as.."
3. Change "Some drugs as" to "Some drugs, such as acetaminophen.."
4. Change "dystyroidism" to "thyroid dysfunction"
5. Change "Despite the wide clinical variety, the histopathology of all PPD is superimposable" to "Despite varying clinical features, all PPD share similar histopathology"
6. Change "may affect all ages but commonly middle-aged to older man, less" to "may affect all ages but commonly occurs in middle-aged to older men, and less"
7. Change "Most of the patients are asymptomatic, someone can complain of weak itching" to " Most patients are asymptomatic, whereas, some complain of mild itching"
8. Change "This disease usually affects middle-aged to older men with a chronic course and occasionally pruritus." to "This disease usually affects middle-aged to older men, has a chronic course, and occasionally presents with pruritus."
9. Change "Differently from other forms, young men are the most affected" to "Unlike other forms, young....."
10. Change "This form is considered an inflammatory ... Schamberg disease, presented for the first time in 1953" to "This form is considered an inflammatory and more extensive variety of Schamberg disease, which was reported for the first time in 1953"
11. Change "It is characterized by usually unilateral linear involvement with lesions similar in appearance to lichen aureus or Schamberg disease" to "It is characterized by lesions that are similar in appearance to lichen aureus or Schamberg disease but are unilateral and linear."
12. Change "Thus, therapeutic abstention must be considered especially in case of asymptomatic patients" to "Thus, conservative or non-pharmacologic management may be considered in the case of asymptomatic patients"
13. Change "Medical strain includes topic and oral drugs, phototherapy and, sometimes, laser-therapy." to "Therapeutic options include topical and oral drugs, phototherapy, and laser-therapy"
14. Change "Continued steroids use should be discouraged for the risk of local atrophy." to "Prolonged steroid use should be discouraged due to the risk of local atrophy"
15. Change "Differently from steroids, calcineurin-inhibitors are not responsible for skin atrophy." to " Unlike steroids, calcineurin-inhibitors...."
16. Consider making a table for the oral medications that have been tried for PPD. Can add columns for the mechanism of action, doses, level of evidence, type of PPD.
17. In the conclusion section: change "thus therapeutic abstention" to "thus non-pharmacological measures"
Author Response
Dear Reviewer 2,
thank you for all the corrections and the suggestions. A table encompassing the most common oral therapies and the dosage has been added.
Kind Regards,
Gianluca Nazzaro